# Differential Alterations of Expression of the Serotoninergic System Genes and Mood-Related Behavior by Consumption of Aspartame or Potassium Acesulfame in Rats

**DOI:** 10.3390/nu16040490

**Published:** 2024-02-08

**Authors:** José Jaime Martínez-Magaña, Alma Delia Genis-Mendoza, Ileana Gallegos-Silva, María Lilia López-Narváez, Isela Esther Juárez-Rojop, Juan C. Diaz-Zagoya, Carlos Alfonso Tovilla-Zárate, Thelma Beatriz González-Castro, Humberto Nicolini, Anayelly Solis-Medina

**Affiliations:** 1Laboratorio de Genómica de Enfermedades Psiquiátricas y Neurodegenerativas, Instituto Nacional de Medicina Genómica, Ciudad de México 14610, Mexico; jimy.10.06@gmail.com (J.J.M.-M.); adgenis@inmegen.gob.mx (A.D.G.-M.); rosa.gallegoss@imss.gob.mx (I.G.-S.); tichovary77@gmail.com (A.S.-M.); 2División Académica de Ciencias de la Salud, Universidad Juárez Autónoma de Tabasco, Villahermosa 86100, Mexico; dralilialonar@yahoo.com.mx (M.L.L.-N.); iselajuarezrojop@hotmail.com (I.E.J.-R.); 3División de Investigación, Departamento de Bioquímica, Facultad de Medicina, Universidad Nacional Autónoma de México, Ciudad de México 04510, Mexico; zagoya@servidor.unam.mx; 4División Académica Multidisciplinaria de Comalcalco, Universidad Juárez Autónoma de Tabasco, Comalcalco 86650, Mexico; thebeagoncas@gmail.com

**Keywords:** aspartame, potassium acesulfame, serotonin receptors, low-nutritional sweeteners, *Htr1a*, *Slc6a4*, *Htr2c*

## Abstract

The use of aspartame (ASP) and potassium acesulfame (ACK) to reduce weight gain is growing; however, contradictory effects in body mass index control and neurobiological alterations resulting from artificial sweeteners consumption have been reported. This study aimed to evaluate the impact of the chronic consumption of ASP and ACK on mood-related behavior and the brain expression of serotonin genes in male Wistar rats. Mood-related behaviors were evaluated using the swim-forced test and defensive burying at two time points: 45 days (juvenile) and 95 days (adult) postweaning. Additionally, the mRNA expression of three serotoninergic genes (*Slc6a4*, *Htr1a*, and *Htr2c*) was measured in the brain areas (prefrontal cortex, hippocampus, and hypothalamus) involved in controlling mood-related behaviors. In terms of mood-related behaviors, rats consuming ACK exhibited anxiety-like behavior only during the juvenile stage. In contrast, rats consuming ASP showed a reduction in depressive-like behavior during the juvenile stage but an increase in the adult stage. The expression of *Slc6a4* mRNA increased in the hippocampus of rats consuming artificial sweeteners during the juvenile stage. In the adult stage, there was an upregulation in the relative expression of *Slc6a4* and *Htr1a* in the hypothalamus, while *Htr2c* expression decreased in the hippocampus of rats consuming ASP. Chronic consumption of ASP and ACK appears to have differential effects during neurodevelopmental stages in mood-related behavior, potentially mediated by alterations in serotoninergic gene expression.

## 1. Introduction

The escalating prevalence of human obesity has been associated with an increase in diet calorie intake, with sugar consumption emerging as a potential factor [1,2,3]. In response to this, various measures focused on reducing this sugar intake have been proposed, including initiatives focused on reducing calories in numerous commercial products, such as candies, sodas, cereals, drink powders, and many other products with high sugar contents. A strategy to achieve this goal involves the substitution of sugar in these products by a group of chemicals known as low-nutritional sweeteners (LNSs) [4]. LNSs possess the notable characteristic of providing the same sweet taste of sugar but with significantly less calorie intake [5].

While LNSs have been associated with a reduction in body mass index (BMI) [6], some investigations suggest a potential inversed effect of these chemical products in BMI control, with reports even implicating young age groups to be the more vulnerable [7,8,9]. A hypothesis attempting to elucidate these contradictory effects in BMI regulation primarily focused on the relationship between sweet taste and satiety signals [10,11]. This conceptual framework states that inconsistent coupling between sweet taste and caloric intake could lead to diminished satiety signals in the brain, thereby contributing to overeating [12]. Concurrently, abnormal feeding behaviors have been linked to mood disorders such as depression and anxiety in both humans [13,14,15] and animal models [16,17,18], suggesting a common signaling mechanism underlying mood, eating behavior, sweet taste, and obesity. In this context, some studies have associated the consumption of LNSs with effects on the neuroendocrine system and mood behaviors [19,20,21]. Notably, one report has highlighted alterations in serotonin levels in mice brains that consume aspartame (a common LNS) [22]. Serotonin, a neurotransmitter, has been implicated in regulating mood and eating behavior, particularly in regulating the consumption of carbohydrate-rich foods [23,24].

Given the potential relationship between LNSs, mood, and obesity, this work aimed to analyze the effects of chronic consumption of two LNSs, aspartame (ASP) and potassium acesulfame (ACK), at two distinct developmental stages (juvenile and adult). Specifically, we sought to investigate changes in weight, mood-related behavior (depression and anxiety), and the differential mRNA expression of specific serotoninergic system genes (*Slc6a4*, *Htr1a*, and *Htr2c*) in male Wistar rats.

## 2. Material and Methods

### 2.1. Ethic Declaration

This present study was revised and approved by Etic Commit in Investigation of the Instituto Nacional de Medicina Genómica (Registry 120/2012/I). The animals were managed according to the Official Mexican Standard, NOM-062-ZOO-1999.

### 2.2. Experimental Design

A total of 48 male Wistar rats were utilized in the experiment provided by the Faculty of Medicine of the National Autonomous University of Mexico, UNAM. They were housed in plastic cages under standard temperature and ventilation conditions, with ad libitum access to food and water. The standard conditions included 12/12-hour cycles light/dark, a relative humidity of 45 to 60%, ventilation cycles of 12 to 15 changes per hour, and a temperature ranging from 18 °C to 26 °C [25].

The rats were divided into six groups. The initial categorization was based on the artificial sweetener consumed, resulting in the following groups: a control group (n = 16), an aspartame-treated group (ASP) (n = 16), and a potassium acesulfame (ACK)-treated group (n = 16). Additionally, these groups were further divided based on the sacrifice day, with rats sacrificed at the postweaning age of 45 days (6.43 weeks) and 90 days (12.86 weeks). This arrangement yielded a total of six groups, each compromising eight animals. A schematic representation of the experiment design is presented in Figure 1.

Sweeteners were administered orally through drinking water, starting on the first day of weaning. The control group received water only, devoid of any additives. The ASP group (ASP from Sigma Chemical Company, Saint Louis, MO, USA) received a concentration of 160 mg/L, while the ACK group (ACK from Sigma Chemical Company, Saint Louis, MO, USA) received a concentration of 240 mg/L, as previously described [25]. Sacrifice was carried out by decapitation. Behavioral tests (defensive burying and forced swing test) were conducted before sacrifice. After sacrifice, the *Slc6a4*, *Htr1a*, and *Htr2c* mRNA expressions were measured in different brain areas (the prefrontal cortex, hippocampus, and hypothalamus). 

Throughout the study of the two stages in the animal models, we sought to better comprehend the pathophysiology of both young and adult animals. Several works have reported that symptoms, the expression of certain genes, and the manifestation of disorders are more observable in the animal adult stage [26]. Consequently, with this scheme, we aimed to observe whether the consumption of sweeteners and the associated behavior and gene expression changes differ between both developmental stages. The expression of serotonergic genes (*Slc6a4*, *Htr1a*, and *Htr2c*) in the brain areas implicated in mood control exhibited variations at different developmental stages, indicating that the neurodevelopmental stages contribute to the divergent effects of ASP and ACK in mood-related behavior.

### 2.3. Weight, Food, and Liquid Measurements

We measured the weight regularly to control the potential impact of ASP and ACK in the modulation of eating behavior and weight gain. Liquid and food consumption were recorded every day throughout the entire experiment. The diet consisted of standard rodent food pellets (DietLab, Land O’Lakes, FL, USA) provided ad libitum. Simultaneously, rat weights were measured weekly, and an average was calculated each week. The measurements began at weaning and continued until the last week before sacrifice.

### 2.4. Behavior Tests

#### 2.4.1. Defensive Burying Test

Polycarbonate cages (34 × 16 × 24 cm) equipped with a 7 cm long electrode were used, positioned on one side of the box and 2 cm above a bed of fine sawdust. Upon contact with the electrode, the animal received an electric shock of 0.3 mA as an adverse stimulus. Instinctively, the animal hides or buries the stimulus, particularly after an encounter. The session was recorded for 10 min, and the accumulated time of burial was considered an anxiety indicator [26].

#### 2.4.2. Forced Swimming Test

The test was conducted following the methodology described by Tang et al. [27]. One week before executing the forced swimming test, the light cycle of the rats was switched. Each rat was placed in a behavioral cylinder (40 cm height, 28 cm diameter) containing 30 cm of water (23 ± 2 °C), preventing them from supporting themselves by touching the bottom with their paws. The test consisted of two segments: an initial pre-test lasting 15 minutes and a second 5-minute test 24 h later. The parameters measured included immobility, swimming, and struggling. Rats were considered immobile when they were floating passively, swimming when actively swimming or making circular movements, and struggling when the rats made active movements with their forepaws to enter and exit the water sideways in the swim chamber. All experiments were recorded on video, and an experienced observer, blind to the experiment design, scored immobility, swimming, and struggling [26,27,28]. In this behavioral test, we utilized the time of swimming, time of floating, and the time of struggling (all measured in seconds) as indicators of depressive-like behaviors.

### 2.5. Relative Gene Expression Quantification

Following decapitation, we extracted the prefrontal cortex, hippocampus, and hypothalamus. Immediately after extraction, the tissues were frozen in liquid nitrogen and stored at −80 °C until processing. Each animal’s frozen brain tissues were homogenized separately. RNA was extracted using the trizol chloroform technique (Life Technologies, Carlsbad, CA, USA). The quantity and quality of the extracted RNA were measured using a Nanodrop2000 spectrophotometer (Thermo Scientific, Waltham, MA, USA) [29]. 

RNA capture and cDNA synthesis were performed using the Maxima First Strand cDNA synthesis for RT-qPCR kit (Thermo Scientific). For mRNA relative expression quantification, three genes were selected: *Slc6a4* (serotonin transporter), *Htr1a* (inhibitory receptor), and *Htr2c* (excitatory receptor). Quantitative real-time PCR (RT-PCR) analyses were performed using primers and probes from TaqMan assays (Applied Biosystems, Waltham, MA, USA). All assays were conducted in triplicates according to the standard cycling protocol. RT-PCR was performed on a 7500 real-time PCR system (Applied Biosystems). The expression rate was normalized with *rs18* gene TaqMan ribosomal RNA control (Applied Biosystems). The delta-delta Ct method was used to measure the relative expression of the genes.

### 2.6. Statistical Analysis

The data are presented as mean and standard deviations. All the data were analyzed for the individual parameters by ANOVA. In cases where a significant “F” test ratio was observed, further analyses were conducted with Tukey’s multiple comparisons or Dunnett’s test, with the significance level set at *p* < 0.05.

## 3. Results

### 3.1. Weight, Food, and Liquid Consumption Measurement

We found no statistically significant differences between the control group and those who drank the LNS (Figure 2). Regarding the food and liquid intake, we found that the rats that consumed artificial sweeteners in their daily water had a higher intake of food and water compared to the control group in all the weeks, and these differences were statistically significant (Figure 2). The higher food and liquid consumption starts at week 1 of the LNS treatment.

### 3.2. Mood-Related Behavior

In the behavioral tests, we found a stage-dependent mood-related behavior alteration. Anxiety-like behavior at the juvenile stages differed at all stages, measured by the time spent burying (F = 4.38, *p* = 0.0211). Nevertheless, in the stratified analysis, only the rats that drank ACK had increased anxiety-like behavior compared to the control group (*p* = 0.0440) (Figure 3). In the adult stages, none of the groups showed differences in anxiety-like behavior.

In the depressive-like behavior, we found differences between groups at the juvenile and adult stages in the swimming time (juvenile: F = 8.06, *p* = 0.0014; adult: F = 4.15, *p* = 0.0253) and immobility time (juvenile: F = 3.55, *p* = 0.0403; adult: F = 4.81, *p* = 0.0161). In the stratified analysis, we found that rats that drank ASP had reduced depressive-like behavior at the juvenile stage, with higher time spent swimming and lower immobility than the control group (Figure 4). Inversely, when we evaluated the rats at the adult stage, rats that drank ASP had an increase in depressive-like behavior, with lower time spent swimming and higher immobility than the control group. In the stratified analysis, the rats that consumed ACK had no differences in depressive-like behavior compared to the control group.

### 3.3. mRNA Expression of Serotonin System Genes (Slc6a4, Htr1a, and Htr2c)

To explore if the consumption of artificial sweeteners changes the expression of genes in the serotoninergic system, we analyzed the mRNA expression of the three serotonin genes: the serotonin transporter (*Slc6a4*), one inhibitory (*Htr1a*), and one excitatory (*Htr2c*) serotonin receptor. We measured the mRNA expression in three brain areas: the prefrontal cortex, hippocampus, and hypothalamus. The prefrontal cortex did not show any changes in the mRNA levels of the analyzed genes. At the juvenile stage, we only found differences in the expression of the mRNA of *Slc6a4* in the hippocampus between rats treated with any LNS and the control group (F = 6.31, *p* = 0.0072) (Figure 5).

In both, the rats that drank ACP and ACK showed a higher expression of *Slc6a4* in the hippocampus compared to the control group. In the adult groups, we found that the mRNA of *Htr2c* was different in the hippocampus between the groups (F = 5.86, *p* = 0.0116). In the stratified analysis, only the group that drank ASP had a lower expression of *Htr2c* than the control group. Meanwhile, the expression of *Slc6a4* and *Htr1a* mRNA was different between the groups in the hypothalamus (*Slc6a4*: F = 12.05, *p* = 0.0006 and *Htr1a*: F = 7.95, *p* = 0.0036) (Figure 5). *Htr1a* and *Slc6a4* mRNA have a higher expression in the groups that consumed artificial sweeteners. No gene showed differential expression in the prefrontal cortex between the groups that consumed artificial sweeteners and the control group.

## 4. Discussion

Our study aimed to analyze the alterations in weight, mood-related behavior, and the expression of serotoninergic genes that modulated appetite (*Slc6a4*, *Htr1a*, and *Htr2c*) in male Wistar rats with chronic consumption of aspartame (ASP) or potassium acesulfame (ACK).

Previous studies reported that low-calorie sweeteners increase body weight in rats [10]. In contrast, our findings showed that the weight of the rats in the groups that drank LNSs did not differ from the control group at any stage. In addition, studies have reported no changes in rats’ body weights by consumption of ACK, and there are even reports of a reduction in mice weight by consumption of ASP [30,31,32]. Recently, Boakes RA et al. [33] found that the increase in body weight was only seen when a tolerance to saccharin exists, possibly explaining the discrepancies in the effect of LNSs on body weight. In our study, rats consuming LNSs had a higher food intake compared to the controls. This elevated energy intake may be an effect of the dysregulation promoted by the LNSs in the perception of sweet taste and caloric content [34]. Swithers S et al. proposed, using rats that consumed saccharin or glucose-sweetened diets, that even when the LNSs have a higher sweet taste, their low-calorie input to the organism emits a signal of calorie restriction and, in consequence, activates to increase overeating [35,36,37]. ASP and ACK are almost 200 times sweeter than sucrose [38]; these LNSs have a higher sense of sweetness but with a lower calorie intake. ASP has a calorie production of 4 kcal/g, and ACK does not provide calories at all [39]. The dysregulation of food energy and sweet taste may be a mechanism underlying the relationship between LNSs and a higher food intake.

Other studies have reported the impact of calorie restriction (i.e., a reduction in calorie intake but consumption of proper protein, vitamins, and water levels) on mood-related behaviors [40,41]. In the behavioral analysis, we found that rats that consumed ASP at juvenile stages had less depression-like behavior, and those that consumed ACK had more anxiety-like behavior. In contrast, rats in the adult stage that consumed ASP had higher depressive-like behavior, and rats that drank ASP had no changes. The differences in the behavior of rats that consumed ASP or ACK may point to different mechanisms of action of these LNSs at the physiological level, which may be a consequence of their different metabolism pathways. ASP is completely metabolized in the intestinal lumen in phenylalanine, aspartic acid, and methanol [42], and ACK is not metabolized at all [43]. This makes ASP unable to circulate in its non-metabolized form but could promote an increase in the concentration of its metabolites in blood circulation. After ASP consumption, plasma phenylalanine increases by 30% [44,45,46]. Neurotransmitters, like serotonin, are synthesized from the essential amino acid (i.e., contributed by diet) tryptophan. It is well known that diseases where an elevation of phenylalanine occurs, like phenylketonuria, reduce the transport of tryptophan to the brain [47] because they share the same neutral amino acid transporters in the blood–brain barrier [48,49]. The increase in circulating phenylalanine promoted by ASP consumption may reduce the transport of tryptophan to the brain (in a concentration-dependent mechanism), which, in consequence, reduces the serotonin synthesis in the serotoninergic neurons [42], affecting the brain process regulated by this neurotransmitter, such as mood and eating behavior [50]. A reduction in serotonin in the mouse brain by consumption of ASP was previously reported by Sharma RP et al. [51]. One potential indicator of a reduction in the brain’s serotonin is the increase in the expression of the serotonin transporter (*Slc6a4*), something we found in the juvenile hippocampus and adult hypothalamus of rats that consumed LNSs, as a possible effect of trying to modulate the deficiency of serotonin. Nevertheless, the question of whether these changes in serotonin synthesis could be translated to changes in the dynamic of serotonin concentration in the synaptic space remains. The reduction in depression-like behavior at the juvenile stage could be an early effect of this serotonin deficiency in rats that consumed ASP; supporting this, a reduction in anxiety-like behavior is one of the main behavioral changes in a serotonin-deficient mouse model (*Tph2* knock-out) [52]. 

Interestingly, ACK also promoted an increase in *Slc6a4* mRNA levels and increased juvenile anxiety behavior. However, this LNS is not metabolized at the gut level and is detected with no chemical modifications in the brain, which could be evidence of a different metabolizing mechanism between ASP and ACK [53]. Also, pointing out this different metabolism-dependent mechanism of ASP and ACK, a reduction in the expression of *Htr2c* was seen in the hippocampus of rats that consumed ASP but not in those that consumed ACK, suggesting that it may be affected by excessive consumption of ASP, in addition to the process regulated by it, such as neuronal and glial reactive processes, or memory [20,54,55]. The *Htr2c* receptor is a key modulator of adult neurogenesis. Song N et al. [56] reported that agonists of *Htr2c* (like the anti-obesity drug lorcaserin) have a direct effect on increasing adulthood neurogenesis survival [57]. A reduction in the mRNA *Htr2c* levels could point to the fact that ASP may reduce neuron survival in the hippocampus by modulating *Htr2c* neurons, something previously reported [25,58]. Even when we did not find mood-related behavioral changes in adult stages by consumption of ACK or ASP, we found an increase in *Slc6a4* and *Htr1a* in the hypothalamus, pointing to a shared mechanism by both LNSs. Further, the hypothalamus participates in appetite regulation [59], principally regulated by genes of the serotonergic system (*Slc6a4* and *Htr1a*). Schellekens et al. [60], in a study with obese mice, found an increase in *Htr1a* in the hypothalamus and a hyperphagic behavior, just like we observed by consumption of both LNSs. Schellekens et al. suggested that the modulation of serotonin genes occurs by a compensatory mechanism during the active food intake phase. Also, these effects are consistent with those promoted by agonists or antagonists of the *Htr1a* receptor [61,62]. Therefore, *Htr1a* and *Slc6a4* overexpression in the hypothalamus may be a mechanism to compensate for the restriction signal emitted by the dysregulation of sweet taste and calorie intake [41], which could reinforce the effect that LNSs have on rats’ body weight changes. Interestingly, in the prefrontal cortex, we did not find differences in the expression of any of the genes studied, which prompted studies in order to clarify which genes are dysregulated in this brain region.

Our study has several limitations. One is the non-inclusion of female rats, which did not allow us to explore sex-dependent changes in behavior and metabolism. Second, we did not measure the amount of other serotonin neurotransmitters to characterize the serotonergic pathway fully.

## 5. Conclusions

Chronic consumption of ASP and ACK could alter depressive- and anxiety-like behaviors in a differential way in the juvenile or adult stage of development in male Wistar rats. Also, the consumption could lead to changes in the expression of some serotoninergic genes (*Slc6a4*, *Htr1a*, and *Htr2c*) in the hippocampus and hypothalamus, which could be the effect of two metabolism-dependent mechanisms of these two LNSs. Nevertheless, more studies still need to clarify the mechanisms of the possible neurological effects of artificial sweeteners.

## Figures and Tables

**Figure 1 nutrients-16-00490-f001:**
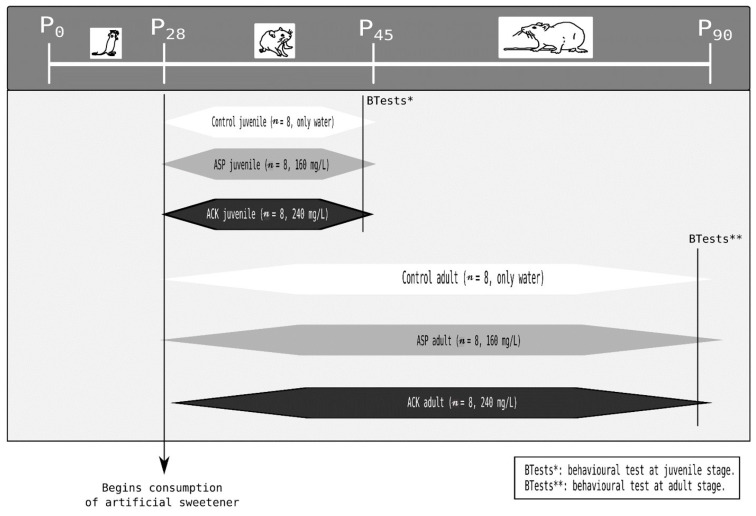
Graphical representation of the experimental design: at the postnatal age of 28 days (P28), in postweaning days, all treated animals consumed the low-nutritional artificial sweeter (LNS) in their daily drinking water. Behavioral tests for juvenile animals were conducted at 45 days (6.43 weeks of LNS treatment). In the adult stage, behavioral tests were performed at 90 days (12.86 weeks of LNS treatment). BTest*: Behavioral test at juvenile stage. BTest**: Behavioral test at adult stage.

**Figure 2 nutrients-16-00490-f002:**
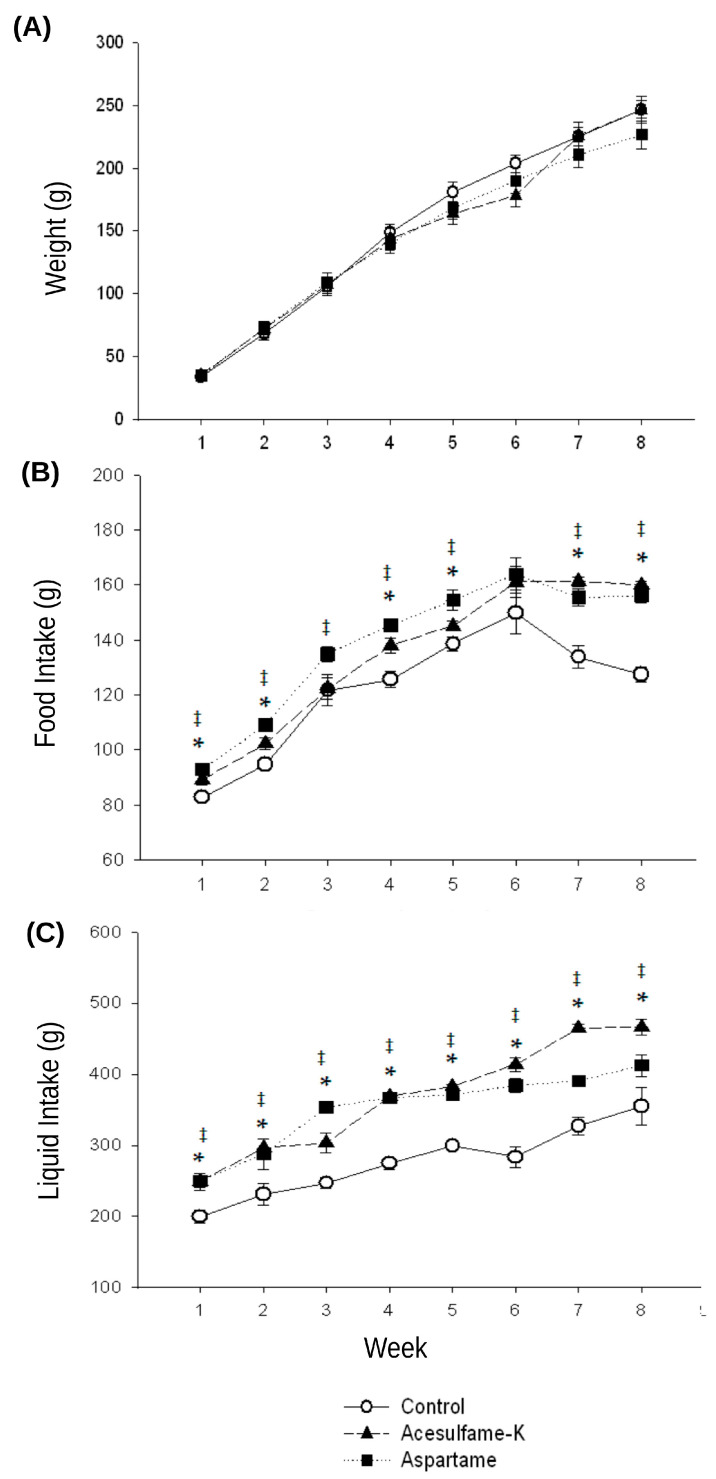
Analysis of the weight, food, and liquid consumption. (**A**) Weekly average weight of rats treated with aspartame (ASP, black squared), acesulfame-K (ACK, black triangle), or control (white circle). No statistically significant changes were seen between the groups. (**B**) Weekly average of food intake. The differences between groups were smaller during the first six weeks of treatment, but at weeks 7 and 8, the difference between the group of rats that consumed LNSs and the control increased. (**C**) Weekly average liquid intake. The rats that consumed LNSs had a higher liquid intake compared to controls. (‡) *p*-value < 0.05 between aspartame and control (*) *p*-value < 0.05 between acesulfame-K and control).

**Figure 3 nutrients-16-00490-f003:**
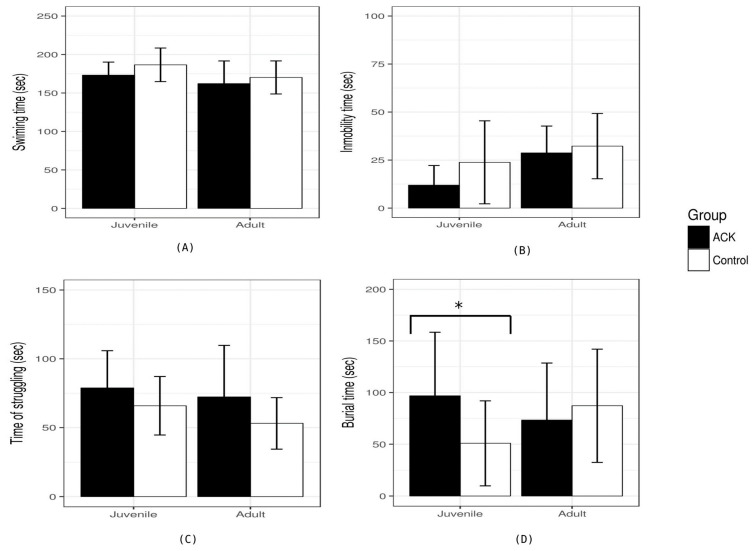
Differences in mood-related behaviors between acesulfame-K (ACK) and control. (**A**) Time spent swimming, and (**B**) immobility time in seconds. (**C**) Time spent struggling in seconds. (**D**) Burial time in seconds. In the juvenile stage (P45), the ACK had a statistically significant higher burial time than the control group. (*) *p*-value < 0.05 between ACK and control.

**Figure 4 nutrients-16-00490-f004:**
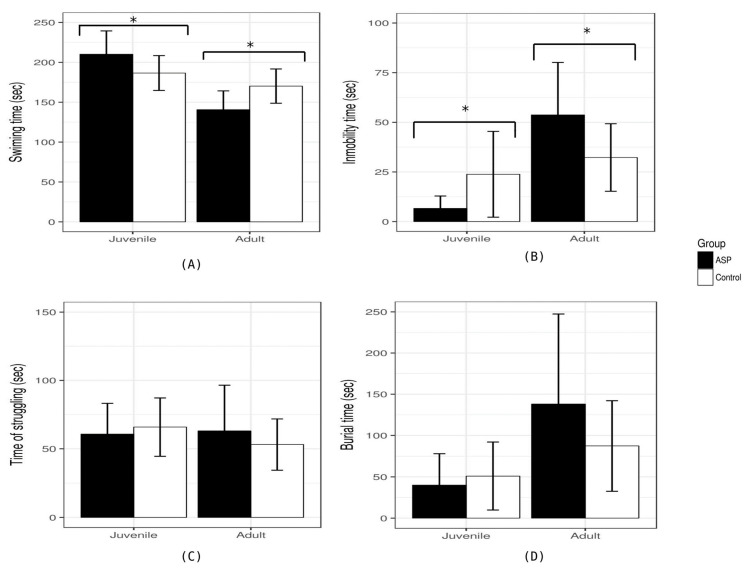
Differences in mood-related behaviors between aspartame (ASP) and control. (**A**) Time spent swimming in seconds. At juvenile stages (P45), the rats that consumed ASP had a higher swimming time reduction in depressive-like behavior. At the difference in the adult stage (P90), the rats that consumed ASP spent less time swimming than the control group, and there was an increase in depressive-like behavior. (**B**) Immobility time in seconds. At juvenile stages (P45), the rats that consumed ASP had a lower immobility time and reduced depressive-like behavior. At a difference in the adult stage (P90), the rats that consumed ASP had higher immobility compared to the control, an increase in depressive-like behavior. (**C**) Time spent struggling in seconds. (**D**) Burial time in seconds. (*) *p*-value < 0.05 between ASP and control.

**Figure 5 nutrients-16-00490-f005:**
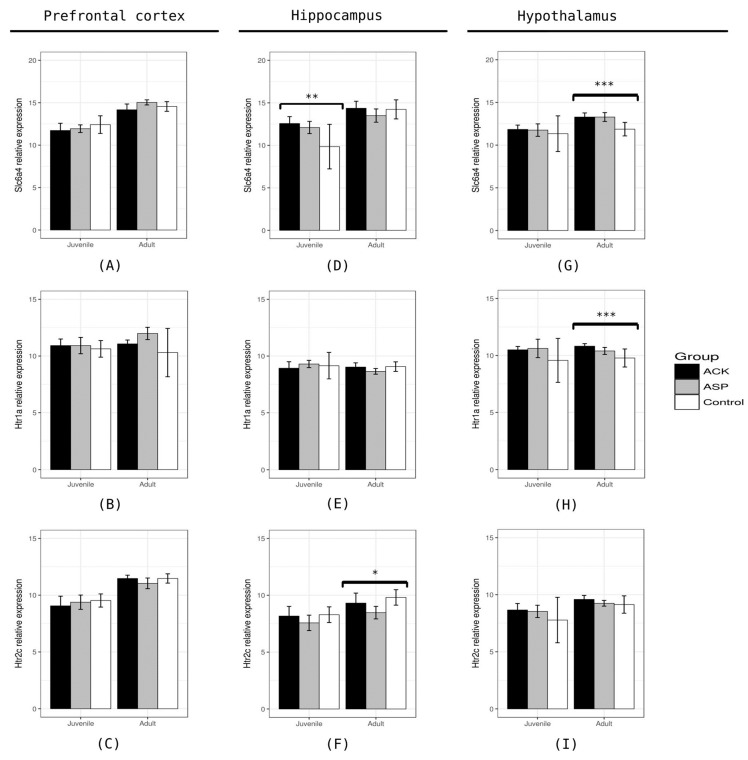
Analysis of mRNA expression levels of serotoninergic genes (*Slc6a4*, *Htr1a*, and *Htr2c*). (**A**–**C**) Relative expression of *Slc6a4*, *Htr1a*, and *Htr2c* in the prefrontal cortex of rats that consumed low-nutritional sweeteners (LNSs) and controls. There were no statistically significant changes in the expression of any serotoninergic genes in the prefrontal cortex in the rats that consumed LNSs at any stage. (**D**–**F**) Relative expression of *Slc6a4*, *Htr1a*, and *Htr2c* in the hippocampus of rats that consumed LNSs. We found an increase in the relative expression of *Slc6a4* in the juvenile stage in the hippocampus of rats that consumed any LNS (**D**). Rats treated with ASP had a reduction in the relative expression of *Htr2c* at the adult stage (**F**). (**G**–**I**) Relative expression of *Slc6a4*, *Htr1a*, and *Htr2c* in the hypothalamus of rats that consumed LNSs. At the adult stage, we found increased *Slc6a4* and *Htr1a* in rats that consumed LNSs (**G**,**H**). (*) *p*-value < 0.05; (**) *p*-value < 0.01; and (***) *p*-value < 0.001.

## Data Availability

The data are available upon request. The data are not publicly available due to privacy reasons.

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
