# Peer review of "Differential Alterations of Expression of the Serotoninergic System Genes and Mood-Related Behavior by Consumption of Aspartame or Potassium Acesulfame in Rats"

_nutrients, 2024, doi:10.3390/nu16040490_

Round 1

Reviewer 1 Report

Comments and Suggestions for Authors

The aim of this study was to assess the impact of chronic consumption of aspartame and potassium acesulfame on mood-related behavior and the expression of serotonin genes in the brains of male Wistar rats. In my opinion, for an original article, a structured abstract, including sections for introduction, objectives, methodology, results, conclusions, and keywords, is more effective, transparent, and convincing for the reader. The research is well-planned, and the manuscript is clear and comprehensible to the reader. The figures excellently aid understanding, and I wouldn't change anything in the manuscript. The results are innovative and thorough, likely to capture the readers' interest. To conclude, you may consider adding a few sentences about limitations that could have influenced the final results. Thank you!

Author Response

Review #1.

The aim of this study was to assess the impact of chronic consumption of aspartame and potassium acesulfame on mood-related behavior and the expression of serotonin genes in the brains of male Wistar rats. In my opinion, for an original article, a structured abstract, including sections for introduction, objectives, methodology, results, conclusions, and keywords, is more effective, transparent, and convincing for the reader. The research is well-planned, and the manuscript is clear and comprehensible to the reader. The figures excellently aid understanding, and I wouldn't change anything in the manuscript. The results are innovative and thorough, likely to capture the readers' interest.

Response: Thank for your kind comments.

To conclude, you may consider adding a few sentences about limitations that could have influenced the final results.

Response: We deeply appreciate the reviewer´s suggestion, and have inserted a sub-section limitations of the study.

Change in the manuscript. Page 10. Lines 321-355.

Some limitations in this study should be noted. One, the no-inclusion of female rat. But, as has been reported, they have differences in behavior and certainly in metabolism. Second, we need to measure the amount of other neurotransmitter, as dopamine and serotonin, to have a complete measurement.

Reviewer 2 Report

Comments and Suggestions for Authors

The present study describes the effects of chronic consumption of two low-nutritional sweeteners such as aspartame and potassium acesulfame on rat models. The authors performed mood-related behavior tests and measurements of specific serotonin genes in different brain areas. I have some questions and comments which should be taken into account for improvement:

Which is the exact nutrition of mice (control and treated)? I read they found a reduction of depression-like behaviour in the forced swim test and those that consumed ACK have an increased in the anxiety-like behaviour in the defensive burial test. Did the authors include a calory restriction protocol?

Could the authors better explain the division of groups between 45 days and 90 days? Which is the biological (functional, emotions, cognition) purpose?

In Figure 3 showing the mRNA expression levels of selected genes, it is not clear between which groups the statistically significant changes are refereed. Please revise the graphs presenting the correct abnormalities between mice groups.

Did the authors assess any neurotransmitters levels such as norepinephrine, serotonin, dopamine or intermediates between brain areas and district time points?

The limitations of the research and the future directions are to be discussed.

Comments on the Quality of English Language

Minor editing of English language required

Author Response

Review #2. The present study describes the effects of chronic consumption of two low-nutritional sweeteners such as aspartame and potassium acesulfame on rat models. The authors performed mood-related behavior tests and measurements of specific serotonin genes in different brain areas. I have some questions and comments which should be taken into account for improvement:

Which is the exact nutrition of mice (control and treated)?

Did the authors include a calory restriction protocol?

Response. Response: We appreciate your comments. In this study, the diet was standard rodent food pellets (DietLab brand) at libitum. No caloric restriction was performed.

Change in the manuscript.

Page 3, Line 104-105. The diet was standard rodent food pellets (DietLab bran) at libitum.

I read they found a reduction of depression-like behaviour in the forced swim test and those that consumed ACK have an increased in the anxiety-like behaviour in the defensive burial test.

Response: We agree with the reviewer, in this study, the ACK group showed a longer immobility time in the forced nothing test and a longer burying time in the defensive burying test.

Could the authors better explain the division of groups between 45 days and 90 days? Which is the biological (functional, emotions, cognition) purpose?

Response: We deeply appreciate the reviewer suggestion. We have added the rationality of the division of groups between 45 days and 90 days.

Change in the manuscript.

Page 2, Line 96 to Page 3, lines 97-103. During the study of animal models, we have sought to understand the pathophysiology of young and adult animals. In several works reported that the symptoms and the expression of some genes and disorders are better observed in adults (Genis-Mendoza et al 2023, 2014). Subsequently, the objective was to observe if the consumption of sweeteners is different in behavior and the expression of genes is different at both ages. The expression of serotonergic genes (Slc6a4, Htr1a and Htr2c) in brain areas involved in mood control varied at different stages of development, indicating that neurodevelopmental stages play a role in the differential effects of ASP and ACK in mood-related behavior.

In Figure 3 showing the mRNA expression levels of selected genes, it is not clear between which groups the statistically significant changes are refereed. Please revise the graphs presenting the correct abnormalities between mice groups.

Response: Thank you for this comment. The figure 3 is Differences in mood-related behaviours between Acesulfame-K (ACK) and control.

Did the authors assess any neurotransmitters levels such as norepinephrine, serotonin, dopamine or intermediates between brain areas and district time points?

The limitations of the research and the future directions are to be discussed.

Response: We have not measured them yet, we have biological material to measure serotonin and dopamine in the future. However, in the present study, we recognized this as a limitation of the study.

Change in the manuscript.

Page 10, Line 321- 324. Some limitations in this study should be noted. One, the no-inclusion of female rat. But, as has been reported, they have differences in behavior and certainly in metabolism. Second, we need to measure the amount of other neurotransmitter, as dopamine and serotonin, to have a complete measurement.

Reviewer 3 Report

Comments and Suggestions for Authors

The study investigated the effects of chronic consumption of ASP and ACK on mood-related behavior and serotonin gene expression in male Wistar rats. The study found that rats consuming ACK showed increased anxiety-like behavior only during the juvenile stage, while those consuming ASP experienced a reduction in depressive-like behavior during the juvenile stage but an increase in the adult stage. The expression of serotoninergic genes (Slc6a4, Htr1a, and Htr2c) in brain areas involved in mood control varied at different developmental stages, indicating that the neurodevelopmental stages play a role in the differential effects of ASP and ACK on mood-related behavior.

Comments:

While the study provides some interesting findings, some limitations should be considered:

1. Using male Wistar rats may not be representative of human responses to sweeteners. Also, gender-specific responses might be overlooked.

2. The study does not account for potential confounding variables such as the rats' overall diet, physical activity, or environmental conditions. These factors could influence the observed alterations in weight and behavior.

3. Conduct a dose-response experiment to examine the effects of different concentrations of aspartame and ACK. This would help establish a dose-dependent relationship and identify potential threshold levels for the observed alterations.

4. The proposed mechanisms linking alterations in serotoninergic genes to changes in behavior and weight are speculative. More detailed molecular and physiological studies are needed to establish a causal relationship.

5. Investigate the metabolism of aspartame and ACK comprehensively, including the identification and quantification of metabolites. This would provide a better understanding of the bioavailability and potential metabolic pathways influencing the observed effects.

6. Behavioral tests like the forced swim test and defensive burial test are subject to interpretation and might not be definitive indicators of mood-related behaviors. Moreover, translating animal behavior to human mood disorders is complex and may not be straightforward.

7. Extrapolating the findings from rats to humans requires caution. Human metabolism and response to sweeteners can differ significantly from that of rats.

Comments on the Quality of English Language

1. The authors need to work on the grammar and sentence structure. There are several lines in the text with grammatical errors and poor sentence structure (lines 19, 44, 49, 51, 52, 64, etc.).

2. The text lacks proper use of tense and verbs. There are a few typographical errors as well. 

Author Response

Review#3

The study investigated the effects of chronic consumption of ASP and ACK on mood-related behavior and serotonin gene expression in male Wistar rats. The study found that rats consuming ACK showed increased anxiety-like behavior only during the juvenile stage, while those consuming ASP experienced a reduction in depressive-like behavior during the juvenile stage but an increase in the adult stage. The expression of serotoninergic genes (Slc6a4, Htr1a, and Htr2c) in brain areas involved in mood control varied at different developmental stages, indicating that the neurodevelopmental stages play a role in the differential effects of ASP and ACK on mood-related behavior.

Comments:

While the study provides some interesting findings, some limitations should be considered:

Using male Wistar rats may not be representative of human responses to sweeteners. Also, gender-specific responses might be overlooked.

Response: That's right, thank you for your comment, the work on male rats was carried out precisely to avoid variations due to sex. But you have to take into account female rats since their behavior is different. Now, recognized this as a limitation in the present study.

Change in the manuscript.

Page 10, Line 321-324. Some limitations in this study should be noted. One, the no-inclusion of female rat. But, as has been reported, they have differences in behavior and certainly in metabolism. Second, we need to measure the amount of other neurotransmitter, as dopamine and serotonin, to have a complete measurement.

The study does not account for potential confounding variables such as the rats' overall diet, physical activity, or environmental conditions. These factors could influence the observed alterations in weight and behavior.

Response: We agree with the reviewer. We want give an explanation. The animals' stay was tried to be as homogeneous as possible, 12 hours of light and 12 hours of darkness. Food and water were weighed per group of rats (5), however it was not possible to measure how much each one ate or drank individually, despite the fact that they were weighed daily. Well, 5 animals were kept per large box, since if they were isolated, it could interfere with behavior.

Conduct a dose-response experiment to examine the effects of different concentrations of aspartame and ACK. This would help establish a dose-dependent relationship and identify potential threshold levels for the observed alterations.

Response: Thank you for your suggestion, we are going to consider doing it.

  1. The proposed mechanisms linking alterations in serotoninergic genes to changes in behavior and weight are speculative. More detailed molecular and physiological studies are needed to establish a causal relationship.

Response. Yes, of course, more molecular studies should be done in this regard, thank you for your comments.

  1. Investigate the metabolism of aspartame and ACK comprehensively, including the identification and quantification of metabolites. This would provide a better understanding of the bioavailability and potential metabolic pathways influencing the observed effects.

Response: We deeply appreciate the reviewer comment. We have not measured them yet, we have biological material, to measure serotonin and dopamine it is contemplated in the future.

  1. Behavioral tests like the forced swim test and defensive burial test are subject to interpretation and might not be definitive indicators of mood-related behaviors. Moreover, translating animal behavior to human mood disorders is complex and may not be straightforward.

Response: We agree. Making the translation between human behavior and rodents can be complicated, however the forced swim and defensive burying tests are standardized tests widely used in research and in the pharmaceutical industry. However, the data should be taken with caution.

  1. Extrapolating the findings from rats to humans requires caution. Human metabolism and response to sweeteners can differ significantly from that of rats.

Response: Thank you for your comments, we agree. The data with animals should be taken with caution.

Comments on the Quality of English Language

  1. The authors need to work on the grammar and sentence structure. There are several lines in the text with grammatical errors and poor sentence structure (lines 19, 44, 49, 51, 52, 64, etc.).
  2. The text lacks proper use of tense and verbs. There are a few typographical errors as well.

Response. We apologize for the mistakes in the manuscript. We also carefully checked and corrected the entire manuscript for typographic, grammatical and formatting errors. Language presentation was improved with assistance from an English speaker with appropriate research background.